# Beyond Unimodal Learning: The Importance of Integrating Multiple Modalities for Lifelong Learning

## Abstract

While humans excel at continual learning (CL), deep neural networks (DNNs) exhibit catastrophic forgetting. A salient feature of the brain that allows effective CL is that it utilizes multiple modalities for learning and inference, which is underexplored in DNNs. Therefore, we study the role and interactions of multiple modalities in mitigating forgetting and introduce a benchmark for multi-modal continual learning. Our findings demonstrate that leveraging multiple views and complementary information from multiple modalities enables the model to learn more accurate and robust representations. This makes the model less vulnerable to modality-specific regularities and considerably mitigates forgetting. Furthermore, we observe that individual modalities exhibit varying degrees of robustness to distribution shift. Finally, we propose a method for integrating and aligning the information from different modalities by utilizing the relational structural similarities between the data points in each modality. Our method sets a strong baseline that enables both single- and multimodal inference. Our study provides a promising case for further exploring the role of multiple modalities in enabling CL and provides a standard benchmark for future research.[1]

## 1 Introduction

Lifelong learning requires the learning agent to continuously adapt to new data while retaining and consolidating previously learned knowledge. This ability is essential for the deployment of deep neural networks (DNNs) in numerous real-world applications. However, one critical issue in enabling continual learning (CL) in DNNs is catastrophic forgetting, whereby the model drastically forgets previously acquired knowledge when required to learn new tasks in sequence (McCloskey & Cohen, 1989). Overcoming catastrophic forgetting is essential to enabling lifelong learning in DNNs and making them suitable for deployment in dynamic and evolving environments.

On the other hand, the human brain excels at CL. A salient feature of the brain that may play a critical role in enhancing its lifelong learning capabilities is that it processes and integrates information from multiple modalities. Various studies have shown that sensory modalities are integrated to facilitate perception and cognition instead of processing them independently (Mroczko-Wasowicz, 2016). In particular, integrating audio and visual information has been shown to lead to a more accurate representation of the environment, which improves perceptual learning and memory (McDonald et al., 2000). Therefore, the multi-modal approach to learning enhances the brain's ability to acquire and consolidate new information.

We hypothesize that integrating multi-modal learning into DNNs can similarly enhance their lifelong learning capability. By combining information from different modalities, the models can develop a more comprehensive understanding of the environment as it receives multiple views of the object, leading to a more accurate and robust representation, which is less sensitive to modality-specific regularities. In recent years, there have been several studies on how to optimally combine multiple modalities, such as vision, audio, and text, and multimodal learning has shown promise in various computer vision applications, including image captioning (Wu et al., 2016) and object

---

[1]The code and dataset will be made publicly available upon acceptance.

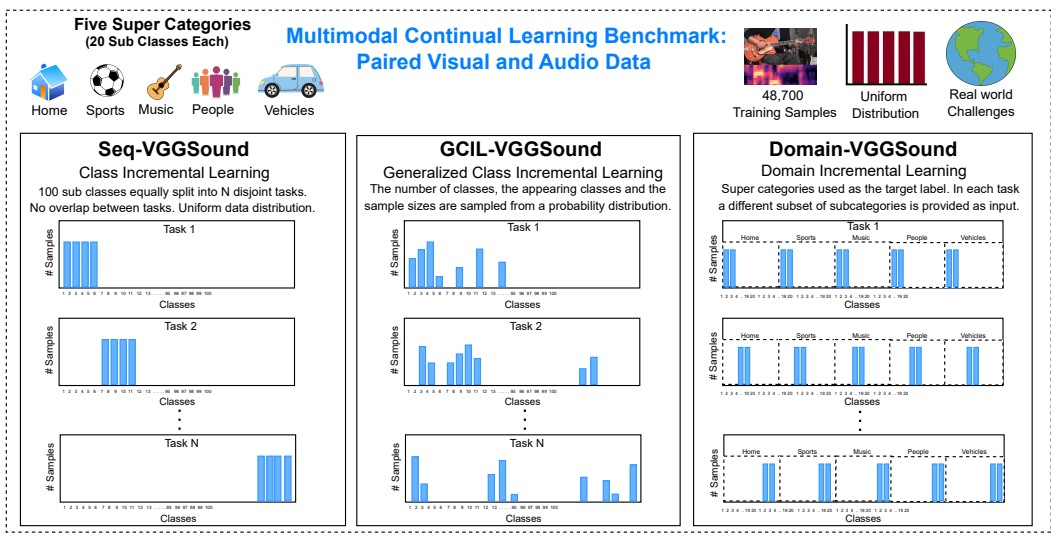

Figure 1: Multimodal continual learning (MMCL) benchmark is based on a subset of the VGGSound dataset (for increased accessibility to the research community), which covers five super categories with twenty sub-classes each. The three CL scenarios cover various challenges that a learning agent has to tackle in the real world and provide correspondence with unimodal benchmarks.

recognition (Karpathy et al., 2014). However, the efficacy of multiple modalities in continual learning and an optimal method for integrating them to mitigate forgetting is understudied, particularly in challenging scenarios such as class incremental and domain incremental learning.

Our study addresses this research gap by studying the role and interactions of multiple modalities in mitigating forgetting in challenging and realistic continual learning settings. Our analysis demonstrates that learning from multiple modalities allows the model to learn more robust and general representations, which are less prone to forgetting and generalize better across tasks compared to learning from single modalities. Furthermore, we show that multimodal learning provides a better trade-off between the stability and plasticity of the model and reduces the bias towards recent tasks. Notably, we argue that in addition to providing complementary information about the task, different modalities exhibit different behavior and sensitivity to shifts in representations depending upon the nature of domain shift, which enhances the stability and transferability of features across the tasks. Therefore, leveraging complementary information from diverse modalities, each exhibiting varying levels of robustness to distribution shifts, can enable the model to learn a more comprehensive and robust representation of the underlying data. The improved representation facilitates improved generalization and retention of knowledge across tasks, thereby enabling effective CL.

Based on the insights from our analysis, we propose a rehearsal-based multimodal CL method that utilizes the relational structural similarities between data points in each modality to integrate and align information from different modalities. Our approach allows the model to learn modality-specific representations from the visual and audio domains, which are then aligned and consolidated in such a manner that a similar relational structure of the data points is maintained in the modality-specific representations as well as the fused representation space. This facilitates the integration and alignment of the two modalities and the subsequent consolidation of knowledge across tasks. Furthermore, our method allows for and enables improved inference with both single and multiple modalities, which significantly improves its applicability in real-world scenarios.

Finally, we introduce a benchmark for class- and domain-incremental learning based on the VGGSound dataset (Chen et al., 2020), which provides a standardized evaluation platform for the community to compare and develop methods for multimodal incremental learning. Our benchmark focuses on challenging scenarios where data is received incrementally over time, simulating real-world scenarios where models need to adapt to new classes or domains without forgetting previously learned information and are exposed to additional challenges including class imbalance, learning over multiple recurrences of objects across tasks, and non-uniform distributions of classes

over tasks. We empirically demonstrate the effectiveness of our approach on this benchmark and provide a strong baseline for future work. Overall, our study presents a compelling case for further exploring multimodal continual learning and provides a platform for the development and benchmarking of more robust and efficient multimodal lifelong learning methods.

## 1.1 RELATED WORK

The various approaches to addressing catastrophic forgetting in continual learning (CL) can be categorized into three main groups. *Regularization-based* methods (Kirkpatrick et al., 2017; Ritter et al., 2018; Zenke et al., 2017; Li & Hoiem, 2017) involve applying regularization techniques to penalize changes in the parameter or functional space of the model. *Dynamic architecture* methods (Yoon et al., 2018; Rusu et al., 2016) expand the network to allocate separate parameters for each task. *Rehearsal-based* methods (Riemer et al., 2018; Arani et al., 2022; Buzzega et al., 2020) mitigate forgetting by maintaining an episodic memory buffer and rehearsing samples from previous tasks. Among these, rehearsal-based methods have proven to be effective in challenging CL scenarios. However, CL has been predominantly studied in an unimodal setting in the visual domain, and the effect and role of multiple modalities in mitigating forgetting in CL is understudied.

On the other hand, there has been substantial attention and progress in learning from multiple modalities (Bayoudh et al., 2021). Various research directions have been explored, depending on specific applications. Some studies have focused on exploring the unsupervised correspondence between multimodal data to learn meaningful representations for downstream tasks (Alwassel et al., 2020; Hu et al., 2019a;b). Furthermore, extensive research is dedicated to leveraging information from multiple modalities to enhance model performance in specific tasks such as action recognition (Gao et al., 2020; Kazakos et al., 2019), audio-visual speech recognition (Hu et al., 2016; Potamianos et al., 2004), visual question answering (Wu et al., 2017; Ilievski & Feng, 2017) and object recognition (Peng et al., 2022). However, these studies focus on generalization in the i.i.d. setting, and progress has not been sufficiently transferred to the CL setting. A recent study Srinivasan et al. (2022) provides a benchmark for multimodal CL; however, they focus on vision and language tasks where each task differs significantly from the other, and their setting does not adhere to the key desiderata outlined in Farquhar & Gal (2018). Furthermore, it lacks correspondence with the challenging and established unimodal CL scenarios (van de Ven & Tolias, 2019), which is essential to assess the benefits of multimodal learning compared to unimodal learning and leveraging the progress in the unimodal CL literature. Our study aims to fill this gap.

## 2 MULTIMODAL CONTINUAL LEARNING BENCHMARK

In order to fully explore the potential and benefits of the increasing amount of multimodal data in real-world applications in enhancing the lifelong learning capability of DNNs, it is imperative to extend the traditional unimodal CL benchmarks to encompass multimodal scenarios. Therefore, we introduce a standardized *Multimodal Continual Learning* (MMCL) benchmark (Figure 1), which aims to simulate challenging and realistic real-world CL scenarios while maintaining correspondence with unimodal CL benchmarks and scenarios (van de Ven & Tolias, 2019).

MMCL benchmark is built upon the VGGSound dataset (Chen et al., 2020), which provides a diverse collection of corresponding audio and visual cues associated with challenging real-world objects and actions and allows for the exploration of multimodal learning scenarios. To ensure accessibility to a wider research community, we select a more uniform subset from the VGGSound dataset, with a total number of samples similar to CIFAR datsets (Krizhevsky et al., 2009) ($\sim$50000 samples uniformly distributed across 100 classes), thus mitigating the requirement for extensive computational resources and memory. We present three distinct CL scenarios within the MMCL benchmark.

**Seq-VGGSound:** This scenario simulates the Class-Incremental Learning (Class-IL) setting, where a subset of 100 classes is uniformly divided into a disjoint set of N tasks. As new classes are introduced in each subsequent task, the learning agent must differentiate not only among classes within the current task but also between classes encountered in earlier tasks. Class-IL evaluates the method's ability to accumulate and consolidate knowledge and transfer acquired knowledge to efficiently learn generalized representations and decision boundaries for all encountered classes.

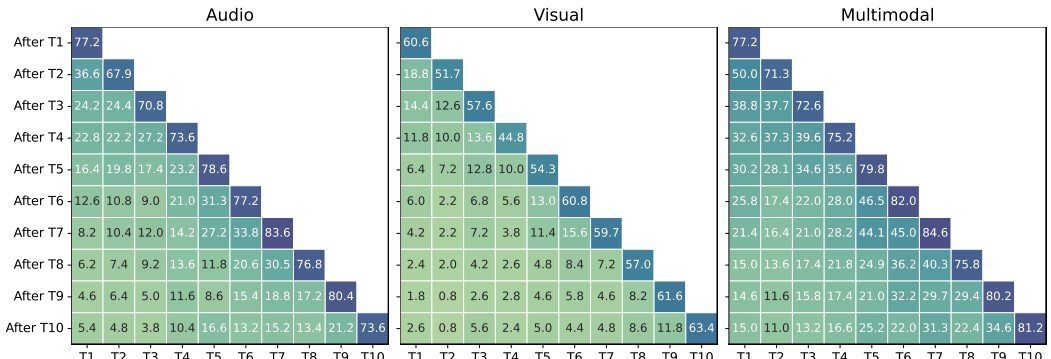

Figure 2: Taskwise performance of models trained with experience replay (1000 buffer size) on multimodal vs. unimodal (audio and visual) data on Seq-VGGSound. As we train on new tasks, T (y-axis), we monitor the performance on earlier trained tasks (x-axis). Multimodal training not only learns the new task better but also retains more performance of earlier tasks.

**Dom-VGGSound:** This scenario simulates the Domain-Incremental Learning (Domain-IL) setting, where the input distribution changes while the output distribution remains the same. To achieve this, we consider the supercategory of classes as the target label, and in each subsequent task, we introduce a new set of sub-classes. We consider five supercategories (Home, Sports, Music, People, and Vehicles). Domain-IL assesses the agent's capability to learn generalized features that are robust to changes in input distribution and to transfer knowledge across different domain shifts.

**GCIL-VGGSound:** This scenario simulates the Generalized Class Incremental Learning setting (GCIL), which captures additional challenges encountered in real-world scenarios where task boundaries are blurry. The learning agent must learn from a continuous stream of data, where classes can reappear and have varying data distributions. In addition to preventing catastrophic forgetting, the CL method must address sample efficiency, imbalanced classes, and efficient knowledge transfer. GCIL-VGGSound introduces assimilated challenges to evaluate the robustness and adaptability of the learning method. For further details on the MMCL benchmark and the settings mentioned above, please refer to Appendix.

## 3 A CASE FOR MOVING BEYOND UNIMODAL CONTINUAL LEARNING

To assess the efficacy of using multiple modalities in CL, we conducted a comprehensive analysis on the Seq-VGGSound scenario, which simulates the challenging Class-IL setting. We aim to investigate the effect of integrating multiple modalities in mitigating forgetting in challenging and realistic CL setting and the different characteristics instilled in the model. To this end, we employ the experience replay (ER) method (with a 1000 buffer size) and compare its performance when learning from unimodal data (Audio and Visual) versus multimodal data. By examining the performance of the models, we aim to understand how multimodal learning influences the model's ability to retain previously learned knowledge while learning new tasks. Our analysis demonstrates notable advantages of learning from multiple modalities over single-modal learning approaches.

### 3.1 IMPROVED CONTINUAL LEARNING PERFORMANCE

The key challenge in CL is mitigating forgetting of earlier tasks as the model learns new tasks. A closer look at the task-wise performances of the models in Figure 2 shows that learning with multimodal data significantly improves both the performance of the model on the current task and performance retention of earlier tasks compared to learning from single modalities. This is further demonstrated in the mean accuracy of the models in Figure 3(a). Interestingly, we observe that different modalities exhibited varying levels of generalization capabilities and susceptibility to forgetting. Notably, the audio domain provides significantly better performance and lower levels of forgetting compared to the visual domain. We argue that different modalities might exhibit different behavior and sensitivity in terms of shifts in representations depending upon the nature of domain

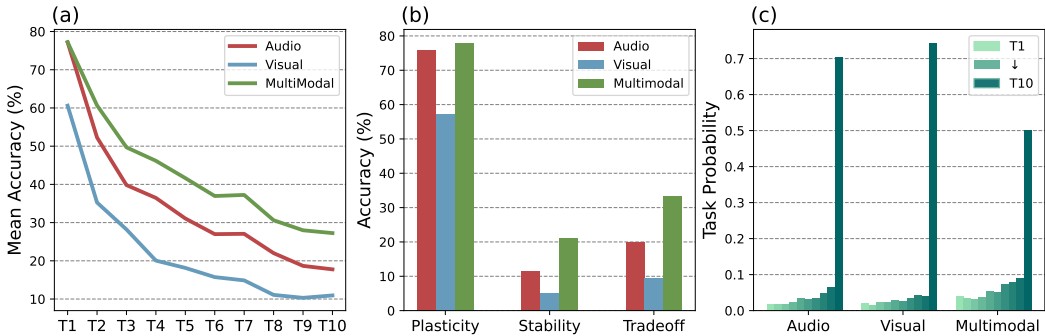

Figure 3: Comparison of experience replay with 1000 buffer size on multimodal data vs. unimodal (audio and visual) data on Seq-VGGSound. (a) provides the mean accuracy of the models on all the tasks (T) seen as training progresses. (b) provides the plasticity stability trade-off of the models while (c) compares the task recency bias. Learning with multimodal data mitigates forgetting, provides a better stability-plasticity trade-off, and reduces the bias toward recent tasks.

shift. For instance, moving from daylight scenarios to nighttime would incur a greater shift in the visual domain compared to audio, whereas moving from an indoor to an outdoor setting may incur a higher shift in the audio domain. Therefore, leveraging the complementary information from different modalities, which shows different degrees of robustness to distribution shifts, can enable the models to capture a more comprehensive and robust representation of the underlying data, leading to improved generalization and retention of knowledge across tasks.

## 3.2 STABILITY PLASTICITY TRADE-OFF

To further investigate the efficacy of multimodal learning in addressing the stability-plasticity dilemma that is central to CL, we follow the analysis in (Sarfraz et al., 2022) to quantify the trade-off between stability and plasticity of models trained with different modalities. Given the task performance matrix $\mathcal{T}$ (Figure 2), stability ($S$) is defined as the average performance of all previous $t-1$ tasks after the learning task $t$, while plasticity ($P$) is quantified by the average performance of the tasks when they are initially learned, computed as mean($\text{Diag}(\mathcal{T})$). Finally, the trade-off is given by $(2 \times S \times P)/(S + P)$.

Stability reflects the model's ability to avoid forgetting and maintain performance on previous tasks, while plasticity captures the model's capacity to learn new tasks. Achieving effective CL requires finding an optimal balance between stability and plasticity. Figure 3(b) shows that multimodal learning considerably improves both the plasticity and stability of the model compared to unimodal training and thus a much better trade-off.

## 3.3 TASK RECENCY BIAS

The sequential learning process in CL introduces a bias in the model towards the most recent task, significantly affecting its performance on earlier tasks (Wu et al., 2019). To investigate the potential of multimodal learning in mitigating the recency bias, we assess the probability of predicting each task at the end of training. To measure the probability, we utilize the softmax output of each sample in the test set and calculate the average probabilities of the classes associated with each task. Figure 3(c) shows that multimodal learning considerably reduces the bias towards the recent task compared to learning from unimodal data. This provides valuable insight into the effectiveness of multimodal learning in addressing performance degradation in earlier tasks and highlights its potential to mitigate the bias inherent in sequential task learning in CL.

Overall, our analysis provides a compelling case for leveraging multiple modalities to mitigate forgetting in CL. The observations of improved CL performance, better stability plasticity trade-off, and reduced recency bias combined with the insights on complementary information and the differential impact of task transitions across modalities underscore the importance of multimodal integration and alignment for effective CL in DNNs.

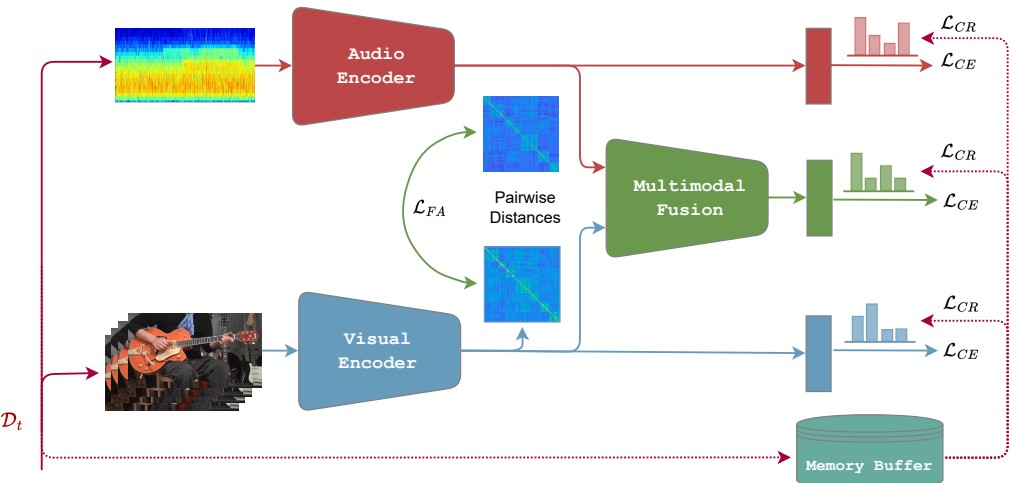

Figure 4: Semantic-Aware Multimodal (SAMM) approach leverages the relational structural information between data points in each modality to integrate and align information from different modalities. Modality-specific representations are learned in the visual and audio encoders, which are then aligned and consolidated to maintain a similar relational structure across the modality-specific representations and the fused representation space. This integration and alignment process enables the consolidation of knowledge across tasks.

## 4 STRUCTURE-AWARE MULTIMODAL CONTINUAL LEARNING

Building upon the insights from our analysis, we present a novel rehearsal-based *semantic aware multimodal* continual learning method, called SAMM, which leverages the relational structure between data points in individual modalities to facilitate the consolidation of individual representations into the combined multimodal representation space. Our approach aims to integrate and align information from different modalities while maintaining a similar relational structure across the modality-specific and fused representation spaces. The goal is to allow the model to learn modality-specific representations from the visual and audio domains while utilizing the inherent structural similarities between data points to align the representations of respective modalities. In the consolidation step, the fused representation space is formed by combining the aligned modality-specific representations. This consolidated representation captures the complementary information from both modalities, resulting in a more comprehensive and robust representation of the underlying data. This enhances knowledge transfer and retention, enabling the model to leverage the combined knowledge learned across tasks. One notable advantage of our method is its versatility and applicability to both single and multiple modalities. By improving the performance of individual modalities and facilitating their integration, our approach can effectively harness the benefits of multimodal learning while accommodating scenarios where only one modality is available or applicable. This flexibility enhances the utility of our method in real-world scenarios where multimodal data may not always be available or one source is noisy or corrupt.

### 4.1 COMPONENTS

Our approach involves training a unified multimodal architecture on a continuous video (paired audio and visual data) stream, $\mathcal{D}$ containing a sequence of $T$ tasks ($\mathcal{D}_1, \mathcal{D}_2, ..., \mathcal{D}_T$). The model comprises an audio encoder $f(.; \theta_a)$ and a visual encoder $f(.; \theta_v)$ followed by a fusion encoder which fuses the audio and visual representations to form the multimodal representations $f(.; \theta_{av})$. Finally, the architecture consists of classification heads for the single modalities ($g(.; \phi_a)$ for audio and $g(.; \phi_v)$ for visual) and multimodal representations ($g(.; \phi_{av})$). For brevity, we represent the audio and visual features as $f_a = f(x_a; \theta_a)$ and $f_v = f(x_v; \theta_v)$ respectively, and the subsequent multimodal features as $f_{av} = f(f_a, f_v; \theta_{av})$. The superscripts $t$ and $b$ indicate features corresponding to task and buffer samples, respectively.

### 4.1.1 MULTIMODAL AND UNIMODAL TASK LOSS

In addition to the supervised loss on the multimodal classifier, we also train the individual modality classification heads. The benefits are two-fold: first, it allows the model to make unimodal inferences, which substantially enhances its applicability in scenarios where either only a single modality is available, or the signal from one modality is corrupted; second, it encourages the task-specific modality to learn richer features and increases the alignment between the feature space of individual modalities, which facilitates the subsequent consolidation in the common multimodal representation space. The supervision loss on the task samples $(x_a^t, x_v^t, y^t) \sim \mathcal{D}_t$ is given by:

$$\mathcal{L}_s^t = \mathcal{L}_{CE}(g(f_{av}^t; \phi_{av}), y^t) + \lambda(\mathcal{L}_{CE}(g(f_a^t; \phi_a), y^t) + \mathcal{L}_{CE}(g(f_v^t; \phi_v), y^t)) \qquad (1)$$

where $\lambda$ (set to 0.01 for all experiments) is the regularization weight that controls the relative weightage between unimodal and multimodal performance. $\mathcal{L}_{CE}$ refers to cross-entropy loss.

### 4.1.2 EXPERIENCE REPLAY

A common and effective approach in CL to mitigate catastrophic forgetting is the replay of samples from previous tasks stored in a small episodic memory buffer $\mathcal{M}$. To maintain the buffer, we employ reservoir sampling (Vitter, 1985), which attempts to match the distribution of the data stream by ensuring that each sample in the data stream has an equal probability of being represented in the buffer and randomly replaces existing memory samples. At any given time, the distribution of the samples in the buffer approximates the distribution of all the samples encountered so far. At each training step, we sample a random batch $(x_a^b, x_v^b, y^b) \sim \mathcal{M}$ from the buffer and apply the supervised classification loss:

$$\mathcal{L}_s^b = \mathcal{L}_{CE}(g(f_{av}^b; \phi_{av}), y^b) + \lambda(\mathcal{L}_{CE}(g(f_a^b; \phi_a), y^b) + \mathcal{L}_{CE}(g(f_v^b; \phi_v), y^b)) \qquad (2)$$

### 4.1.3 CONSISTENCY REGULARIZATION

While the replay of samples from previous tasks can help alleviate catastrophic forgetting, it struggles to accurately approximate the joint distribution of all tasks encountered thus far, particularly when using smaller buffer sizes. To address this limitation, additional information from the model's earlier state is necessary to maintain the parameters closer to their optimal minima for previous tasks. Similar to earlier works (Buzzega et al., 2020; Arani et al., 2022), we employ consistency regularization on the model's logit response, which encodes valuable semantic information about the representations and decision boundaries, thereby facilitating knowledge retention.

In addition to storing and replaying samples from previous tasks, our approach involves saving the unimodal and multimodal output logits $(z_a, z_v, z_{av})$ of the model. The inclusion of unimodal and multimodal outputs in the consistency regularization process further strengthens the retention of semantic relations. Furthermore, to encourage consistency across the modalities and improve the quality of supervision, we employ dynamic consistency whereby for each sample, we select reference logits $z_r$ from the modality that provides the highest softmax score for the ground truth class. By encouraging the model to maintain consistent responses across modalities, we facilitate the integration and alignment of information from different modalities, enhancing the overall performance of the model in multimodal CL scenarios. The consistency regularization loss is given by:

$$\mathcal{L}_{cr}^b = \mathcal{L}_{MSE}(g(f_{av}^b; \phi_{av}), z_r^b) + \lambda(\mathcal{L}_{MSE}(g(f_a^b; \phi_a), z_r^b) + \mathcal{L}_{MSE}(g(f_v^b; \phi_v), z_r^b)) \qquad (3)$$

where $\mathcal{L}_{MSE}$ refers to the mean squared error loss.

### 4.1.4 SEMANTIC-AWARE FEATURE ALIGNMENT

One of the key challenges in multimodal learning is the alignment of feature representations across different modalities. This alignment is crucial for the effective consolidation of knowledge in a shared multimodal representation space. We address this challenge by leveraging the relational structure between data points in each modality to integrate and align information from different modalities. Our method allows the model to learn modality-specific representations from both the visual and the audio domains. These modality-specific representations are then aligned and consolidated in a way that preserves a similar relational structure among the data points within each

Table 1: Comparison of different methods on individual and multiple modalities on different CL scenarios based on VGGSound dataset. We report mean and 1 s.t.d of three seeds

| Buffer | Method | Seq-VGGSound | | | Dom-VGGSound | | |
|---|---|---|---|---|---|---|---|
| | | Audio | Visual | Multimodal | Audio | Visual | Multimodal |
| | JOINT | $53.47_{\pm1.62}$ | $34.52_{\pm0.28}$ | $58.21_{\pm0.24}$ | $57.48_{\pm0.80}$ | $42.87_{\pm2.04}$ | $61.66_{\pm1.40}$ |
| | SGD | $7.60_{\pm0.33}$ | $6.37_{\pm0.33}$ | $8.24_{\pm0.09}$ | $26.89_{\pm0.17}$ | $24.80_{\pm0.12}$ | $27.38_{\pm0.35}$ |
| 500 | ER | $13.92_{\pm1.07}$ | $9.07_{\pm0.39}$ | $21.44_{\pm1.76}$ | $31.31_{\pm0.80}$ | $27.36_{\pm1.60}$ | $31.85_{\pm1.09}$ |
| | SAMM | $23.61_{\pm1.06}$ | $8.90_{\pm0.35}$ | $\mathbf{26.34_{\pm0.42}}$ | $\mathbf{36.27_{\pm0.29}}$ | $24.98_{\pm0.41}$ | $35.74_{\pm0.59}$ |
| 1000 | ER | $18.06_{\pm0.44}$ | $11.23_{\pm0.57}$ | $28.09_{\pm0.77}$ | $35.31_{\pm0.65}$ | $27.73_{\pm0.99}$ | $36.00_{\pm1.08}$ |
| | SAMM | $28.59_{\pm0.77}$ | $10.08_{\pm0.34}$ | $\mathbf{34.51_{\pm2.37}}$ | $38.63_{\pm0.43}$ | $26.53_{\pm0.12}$ | $\mathbf{39.49_{\pm0.36}}$ |
| 2000 | ER | $23.41_{\pm0.50}$ | $14.19_{\pm0.32}$ | $34.02_{\pm0.40}$ | $38.23_{\pm0.72}$ | $29.28_{\pm0.63}$ | $39.30_{\pm1.55}$ |
| | SAMM | $32.20_{\pm0.28}$ | $11.60_{\pm0.43}$ | $\mathbf{37.76_{\pm2.94}}$ | $42.53_{\pm0.47}$ | $28.12_{\pm0.31}$ | $\mathbf{43.72_{\pm0.34}}$ |

modality. This encourages the learned multimodal representations to capture the essential relationships and similarities between modalities and promotes a holistic understanding of the underlying data, improving the ability of the model to learn and transfer knowledge across tasks.

To this end, we employ the distance-wise relation knowledge distillation loss from Park et al. (2019) separately on the task samples and the buffer samples:

$$\mathcal{L}_{FA}^{b,t} = \sum_{(x^i, x^j) \in \chi_{b,t}^2} \mathcal{L}_H(\psi_D(f_a^i, f_a^j), \psi_D(f_v^i, f_v^j)), \qquad \psi_D(f^i, f^j) = \frac{1}{\mu}||f^i - f^j||_2 \qquad (4)$$

where $\mathcal{L}_H$ is Huber loss (Huber, 1992), $\mu$ represents the mean distance between all pairs within the given batch of $\chi_{b,t}^2$, whether it is from the buffer or task samples. Essentially, the loss encourages a similar pair-wise relationship structure in the different modalities by penalizing distance differences between their individual output representation spaces.

Finally, we combine individual losses to train the multimodal architecture.

$$\mathcal{L} = \mathcal{L}_s^t + \mathcal{L}_s^b + \beta \cdot \mathcal{L}_{cr}^b + (\mathcal{L}_{FA}^t + \mathcal{L}_{FA}^b) \qquad (5)$$

where $\beta$ controls the strength of consistency regularization loss.

### 4.1.5 DYNAMIC MULTIMODAL INFERENCE

While utilizing multiple modalities improves generalization, for each sample, ideally the model should be able to weigh each modality based on how informative it is. This allows us to deal with scenarios where one modality is corrupted, noisy, or occluded. To this end, we use a weighted ensemble of the classifiers based on the softmax confidence score:

$$z_o = \max(\sigma(z_a)) \cdot z_a + \max(\sigma(z_v)) \cdot z_v + \max(\sigma(z_{av})) \cdot z_{av} \qquad (6)$$

For such a weighting scheme to work well, the model should be well-calibrated so that the confidence score is a good proxy of the modality's performance. At the end of each task, we calibrate the classifiers using temperature scaling (Guo et al., 2017) on the buffer samples. This provides us with a simple and effective approach for leveraging different modalities based on their quality of signal.

## 5 EMPIRICAL EVALUATION

We compare our semantic-aware multimodal learning approach with the baseline Experience Replay (ER) (Riemer et al., 2018) method under uniform experimental conditions (Section A.2) on a wide range of multimodal CL scenarios that cover various challenges that a lifelong learning agent has to tackle in the real world. Note that the baseline ER method requires separate training on each individual modality and can make inferences on either multimodal data or on individual modalities. In contrast, our proposed approach utilizes a unified architecture that is able to make inferences

Table 2: Performance comparison on individual and multiple modalities on GCIL-VGGSound.

| Buffer | Method | Uniform | | | Longtail | | |
|---|---|---|---|---|---|---|---|
| | | Audio | Video | MultiModal | Audio | Video | MultiModal |
| – | JOINT | $44.02_{\pm 1.51}$ | $26.23_{\pm 0.63}$ | $49.32_{\pm 0.43}$ | $43.23_{\pm 1.38}$ | $25.19_{\pm 0.76}$ | $47.17_{\pm 0.31}$ |
| | SGD | $20.34_{\pm 0.51}$ | $11.47_{\pm 0.79}$ | $24.73_{\pm 0.40}$ | $19.00_{\pm 0.43}$ | $10.43_{\pm 0.67}$ | $22.03_{\pm 0.67}$ |
| 500 | ER | $24.57_{\pm 0.44}$ | $13.80_{\pm 0.53}$ | $29.76_{\pm 0.82}$ | $24.30_{\pm 0.33}$ | $12.81_{\pm 0.11}$ | $28.58_{\pm 0.73}$ |
| | SAMM | $27.41_{\pm 0.41}$ | $11.90_{\pm 1.65}$ | $\mathbf{34.34}_{\pm 0.78}$ | $27.25_{\pm 0.65}$ | $12.06_{\pm 0.22}$ | $\mathbf{34.16}_{\pm 0.81}$ |
| 1000 | ER | $27.32_{\pm 0.38}$ | $15.53_{\pm 0.30}$ | $34.27_{\pm 0.77}$ | $27.25_{\pm 0.93}$ | $14.24_{\pm 0.25}$ | $31.60_{\pm 0.94}$ |
| | SAMM | $29.99_{\pm 0.41}$ | $13.18_{\pm 0.64}$ | $\mathbf{38.04}_{\pm 1.51}$ | $28.52_{\pm 0.40}$ | $12.64_{\pm 0.12}$ | $\mathbf{36.15}_{\pm 0.30}$ |
| 2000 | ER | $31.30_{\pm 0.28}$ | $17.25_{\pm 0.09}$ | $37.77_{\pm 0.80}$ | $29.75_{\pm 0.46}$ | $17.31_{\pm 0.21}$ | $35.66_{\pm 0.53}$ |
| | SAMM | $31.96_{\pm 0.76}$ | $14.35_{\pm 0.58}$ | $\mathbf{42.08}_{\pm 1.89}$ | $30.13_{\pm 0.68}$ | $13.09_{\pm 0.57}$ | $\mathbf{40.33}_{\pm 0.38}$ |

under a multimodal setting, as well as individual modalities, enabling more efficient training and inference while enhancing the applicability of the method.

Table 1 shows that SAMM improves the performance of the model in the majority of the scenarios. The improvement under Seq-VGGSound shows that SAMM effectively mitigates forgetting and can learn well-aligned general representations that are able to transfer knowledge across tasks. In particular, the considerable gains over multimodal ER show that our proposed structure-aware multimodal learning approach is able to better leverage the complementary information in different modalities to learn a more robust representation. Interestingly, we observe an order of magnitude gains in audio even compared to a model trained specifically on it. We argue that multimodal learning leads to richer feature representations that are more generalizable, and aligning the two modalities not only facilitates the consolidation in the multimodal representation space but also allows efficient knowledge transfer between modalities. Note that visual generally performs much worse compared to audio as for many classes (e.g. clapping) audio cues are more informative and visual cues (e.g. performers on stage) can be misleading. We observe similar gains in the Domain-VGGSound setting, where the model is required to learn robust generalizable features, which are robust to input distribution. Dom-VGGSound exposes the model to sharp input distribution shift as the subclasses change at the task transition, and hence the competitive performance of our approach in this setting suggests that it is able to learn generalizable features that are more robust to distribution shifts.

To further evaluate the versatility of our approach, we also consider GCIL, which introduces additional complexities of real-world scenarios where tasks are nonuniform, and classes can reoccur with different distributions. Table 2 compares the performance of our method with ER under the uniform and longtail data distribution settings. Consistent gains with our method show the effectiveness of our approach in learning across multiple occurrences of the object, improving the sample efficiency, and enhancing the robustness to imbalanced data. We attribute the performance gains in our method to the efficient utilization of individual modalities by encouraging them to learn discriminative features using the modality-specific classification head and aligning them using the semantic aware feature alignment loss, which facilitates the consolidation of modalities in a common representation space. Overall, our results provide strong motivation for further exploration of multimodal learning in CL and developing methods that efficiently utilize the complementary information in different modalities and varying robustness to distribution shifts at the task transition to enable efficient CL.

## 6 CONCLUSION

Our study highlights the potential of multimodal continual learning in mitigating forgetting in DNNs. By integrating audio and visual modalities, we observed reduced forgetting and enhanced adaptability. Additionally, our analysis showed a better trade-off between the stability and plasticity of the model and reduced bias toward recent tasks. We proposed a rehearsal-based multimodal CL method that aligns and consolidates modality-specific representations, achieving effective knowledge transfer across tasks. Our benchmark based on the VGGSound dataset provides a standardized evaluation platform, and our method serves as a strong baseline for future research. We encourage the further exploration of multimodal CL to develop robust models for dynamic environments.

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

# A APPENDIX

## A.1 ABLATION STUDY

In order to gain a deeper understanding of the individual contributions made by each component of the SAMM, we systematically introduced each component one at a time and evaluated its impact on the model's performance across both unimodal and multimodal inference. Table 3 shows that all the components (Unimodal classifiers (UM) + Consistency Regularization (CR), Feature Alignment (FA), and Dynamic Inference (DI)) contribute towards the final performance of the method.

Unimodal classifiers coupled with consistency regularization demonstrate a substantial positive impact on the performance of single modalities and also show improvements in multimodal scenarios. Furthermore, the incorporation of feature alignment leads to enhanced performance in both the audio and audio-visual domains. The improvement in the multimodal scenario shows that the alignment of the individual modalities facilitates the consolidation of complementary information in the fused representation space. However, perhaps due to the nature of the datasets (where the visual domain is sometimes unrelated to the audio cues e.g. clapping of the audience in the audio domain while visual shows the performers taking a standing ovation), this alignment and fusion can lead to a decrement in the visual modality performance. Additionally, dynamic inference is specifically designed to leverage maximum knowledge from both unimodalities, and our experiments confirm its effectiveness, resulting in an approximately 10% improvement in the performance of the multimodal system.

Table 3: Contribution of the different components (UM: Unimodal classifiers, CR: Consistency Regularization, FA: Feature Alignment, and DI: Dynamic Inference) of SAMM on the performance of the model on Seq-VGGSound with 1000 buffer size.

| $UM+CR$ | $FA$ | $DI$ | Audio | Video | Multimodal |
|:---:|:---:|:---:|:---:|:---:|:---:|
| ✗ | ✗ | ✗ | $1.05_{\pm 0.03}$ | $1.25_{\pm 0.06}$ | $28.09_{\pm 0.77}$ |
| ✓ | ✗ | ✗ | $24.03_{\pm 1.08}$ | $10.32_{\pm 0.28}$ | $29.13_{\pm 0.89}$ |
| ✓ | ✓ | ✗ | $28.59_{\pm 0.77}$ | $10.08_{\pm 0.34}$ | $31.48_{\pm 1.07}$ |
| ✓ | ✓ | ✓ | $28.59_{\pm 0.77}$ | $10.08_{\pm 0.34}$ | $34.51_{\pm 2.37}$ |

## A.2 EXPERIMENTAL SETUP

For all our multimodal experiments, we follow the setup in (Peng et al., 2022) and employ ResNet18 (He et al., 2016) architecture as the backbone. The visual encoder takes multiple frames as input, following Peng et al. (2022); Zhao et al. (2018), while the audio encoder modifies the input channel of ResNet18 from 3 to 1, as done in Chen et al. (2020). The videos in VGGSound dataset have a duration of 10 seconds, and we extract frames at a rate of 1 frame per second and uniformly sample 4 frames from each 10-second video clip as visual inputs in the temporal order. For audio data processing, following Peng et al. (2022), we transform the entire audio into a spectrogram with dimensions of $257 \times 1{,}004$ using the librosa (McFee et al., 2015) library. We employ a window length of 512 and an overlap of 353. To optimize our model, we use stochastic gradient descent (SGD) with a learning rate of 0.1. From the architecture perspective, we adapt the ResNet architecture (following Peng et al. (2022); Zhao et al. (2018)) for the visual encoder and audio encoder. The first conv layer of both audio and visual encoder uses a kernel size of 7 and stride 2 followed by max pooling with kernel 3 and stride 2. For the Multimodal fusion, we use FiLM Perez et al. (2018) to fuse the representations from audio and video encoder which applies affine transformations to the intermediate representations to learn how to combine them effectively. The fused representations are passed to the multimodal classifier whereas the individual representations are flattened and passed to modality-specific representations. This allows the model the flexibility to perform both unimodal and multimodal predictions. The hyperparameters for our approach are set using a small validation set. For all our experiments we report the mean and 1 std of three different seeds.

## A.3 HYPERPARAMETERS

As the goal of our study was to understand the role and benefits of utilizing multiple modalities in CL and not to extract the best possible results and establish a state-of-the-art, we did not conduct an

extensive hyperparameter search for different buffer sizes and settings. For all our experiments, we set $\lambda$ to 0.01. For Dom-VGGSound, we use $\beta$=0.1 and for Seq-VGGSound and GCIL-VGGSound, we used $\beta$=1. Note that the results can be improved with hyperparameter tuning.

### A.4 MMCL BENCHMARK DETAILS

We designed the *multimodal Continual Learning (MMCL)* benchmark with the following principles: (1) Adherence to the desiredata's in Farquhar & Gal (2018); (2) Correspondence with unimodal benchmarks; (3) Assessment of challenges that a learning agent encounters in the real world; and (4) Accessibility to the wider research community.

Therefore, we considered the challenging Class-IL, Domain-IL, and Generalized-Class-IL (GCIL) scenarios, each of which simulates different sets of challenges for a learning agent in our dynamic and complex environment. To make the benchmark accessible to the wider research community and to facilitate further development of multimodal CL methods, we kept the overall dataset size and the number of classes similar to the widely adopted CIFAR-100 dataset. This ensures that the benchmark does not require excessively intensive computational and memory resources for experimentation.

We selected a subset of 5 supercategories (animals, music, people, sports, and vehicles) from the VGG-Sound dataset, each containing 20 subclasses (see Figure 5). This resulted in an overall class count of 100, similar to CIFAR-100. As with CIFAR-100, we aimed to have 500 training samples and 50 test samples for each class. However, due to the distribution of classes in the original VGG-Sound dataset and the current unavailability of some YouTube videos, it was not possible to acquire 500 samples for all the classes. Nevertheless, for the vast majority of classes, our benchmark is based on a uniform set of 500 training samples (see Figure 5). Notably, for the 'sports' supercategory, we have a lower number of samples for the classes.

Figure 5 provides details about the selected subclasses within each of the 5 supercategories and their respective training sample counts. For **Seq-VGGSound**, we randomly shuffled the classes and divided the dataset into 10 disjoint tasks, each containing 10 classes (the order of classes is each supercategory is provided in Figure 5). In **Dom-VGGSound**, we assigned the supercategory as the target label and created 10 tasks, with each task consisting of samples from the next two subclasses in the order presented in Figure 5. For example, in Task 1, we utilized samples from the 'owl hooting' and 'cricket chirping' subclasses for the 'animals' supercategory, and in Task 2, we used 'gibbon howling' and 'woodpecker pecking tree,' and so forth. In the case of **GCIL-VGGSound**, we followed settings similar to Mi et al. (2020): 20 tasks, each with a total of 1000 samples, and a maximum of 50 classes in each task. For reproducibility, we fixed the GCIL seed at 1992, given the probabilistic nature of GCIL. For further reproducibility and to make the dataset available to the research community at large, the dataset files and code for the method and benchmark will be made publicly available upon acceptance.

### A.5 ACTIVATION MAPS

To gain a better understanding of the effect of multimodal learning, we look at the GradCam Selvaraju et al. (2017) activations of the model. We use the gradients of the first convolution layer of the visual net with respect to the output logit corresponding to the class label. The activation maps in Figure 6 show that MultiModal ER allows the model to attend to regions in the image that are associated with the label and localize the sound. SAMM considerably improves sound localization and attends to the most pertinent regions in the image. For playing saxophone, SAMM attends more to the saxophone and the mouth regions. Similarly, for tap dancing, SAMM rightfully attends more to the legs. This shows that multiple modalities and structure-aware alignment in SAMM enable the model to learn more holistic representations and focus on the regions associated with the class. The enhanced localization of sound in SAMM shows that our method effectively aligns the two modalities and learn a better representation.

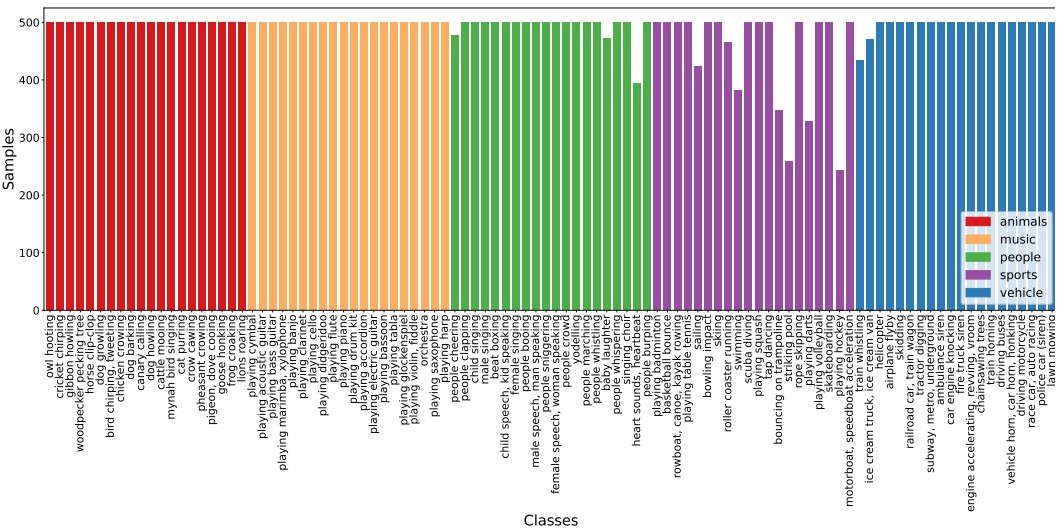

Figure 5: The distribution of training samples count for the subclasses in each supercategory in the MMCL benchmark. We consider 5 supercategories with 20 subclasses each and aim for a uniform sample count of 500 training samples and 50 test samples.

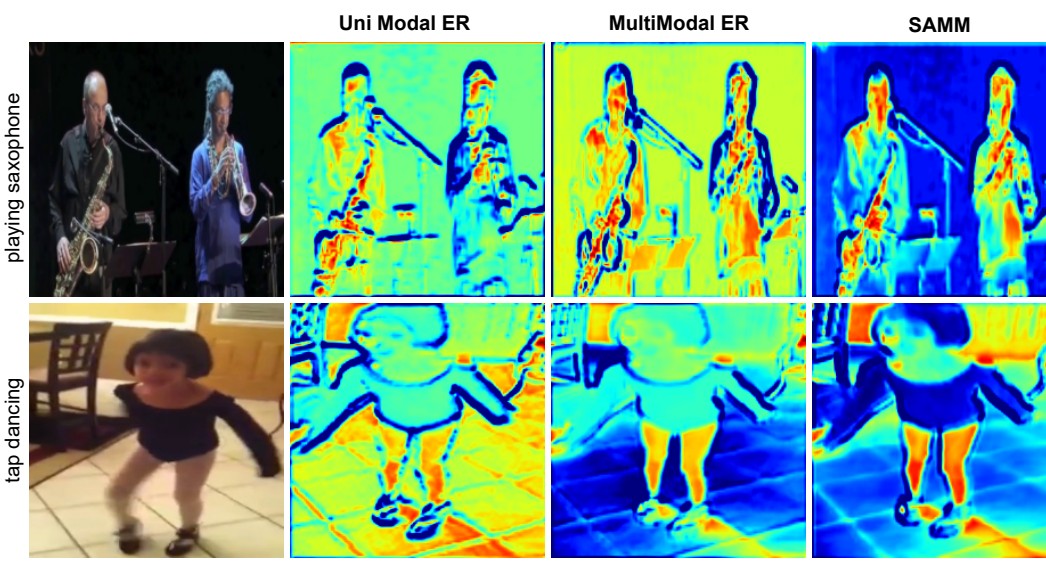

Figure 6: Gradcam activations for Visual ER, Multimodal ER, and SAMM. Multimodal ER attends more to the regions that are associated with the class labels and localizes the sound regions better. This is considerably improved with SAMM which shows that it effectively aligns the two modalities.

