# OpenReview forum: "Beyond Unimodal Learning: The Importance of Integrating Multiple Modalities for Lifelong Learning"
_ICLR.cc/2024/Conference — Submitted to ICLR 2024_

### Official Review · Reviewer_eqa5 · 2023-10-29

**Soundness:** 3 good
**Presentation:** 3 good
**Contribution:** 2 fair
**Rating:** 5
**Confidence:** 3

**Summary:**

This paper studies an under-explored problem --- leveraging multiple modalities for lifelong learning. Towards this end, the authors (1) provide a benchmark for this task, sourced from VGGSound; (2) conduct a case study demonstrating the advantages of using multiple modalities over a single modality; (3) develop an approach to leverage relational structural information in each modality for better integration of multimodal information.

**Strengths:**

+ The proposed benchmark covers three CL scenarios and can be beneficial to the community.
+ The analysis in Section 3 makes sense and provides empirical evidence for the superiority of multiple modalities over single modality in CL.
+ The paper has good motivation and is well organized.

**Weaknesses:**

My major concern with this paper is the lack of comparison and experiments. The evaluation seems a bit weak to me as all the experiments are conducted on VGGSound only, and the baseline Experience Replay for comparison with the proposed approach is from 2018. I wonder if the authors could apply some more recent unimodal CL approaches ([1][2] etc.) to the problem.

[1] SS-IL: separated softmax for incremental learning
[2] Class-incremental learning by knowledge distillation with adaptive feature consolidation.

---
Also, in terms of comparison with multimodal CL approaches:
+ (1) Could the authors further clarify why the proposed approach can not be applied to vision-language? What is the advantage of the proposed benchmark compared with [3], besides the modality difference?
+ (2) I understand that [4] is published after the submission ddl, but it would be good if the authors could comment a few sentences about the differences with them in the rebuttal.

[3] Climb: A continual learning benchmark for vision-and-language tasks.
[4] Audio-Visual Class-Incremental Learning

---
For the semantic-aware feature alignment, I wonder if the authors can provide some visualization examples to demonstrate that the model indeed learns the desired modality-specific features, such as Figure 4 in [5].

[5] The Modality Focusing Hypothesis: Towards Understanding Crossmodal Knowledge Distillation

---
Typo, Figure 4 caption, "leverage leverages"

**Questions:**

See weaknesses.

---

> ### Author Response · Authors · 2023-11-21
> **Author's response (1/2)**
>
> We thank the reviewer for the positive feedback and insightful comments. We are glad that the reviewer sees the benefits of multimodal learning and its value in the proposed benchmark for the research community.
>
> >My major concern with this paper is the lack of comparison and experiments. The evaluation seems a bit weak to me as all the experiments are conducted on VGGSound only, and the baseline Experience Replay for comparison with the proposed approach is from 2018.
>
> we would like to underscore that our study's objective was not to propose a state-of-the-art solution but rather to present a compelling case for advancing toward multimodal CL and establishing a standardized benchmark for multimodal CL to encourage research in this promising direction. The primary contribution of our work lies in demonstrating the performance gains of Multimodal Experience Replay compared to unimodal experience replay, particularly in challenging CL scenarios, corresponding to the established CL settings in the unimodal context. Additionally, we introduced a promising approach for aligning modalities to enhance complementary learning, facilitating both multimodal and unimodal inference as an initial baseline for future research to build upon.
>
> Our choice of Experience Replay in this study was motivated by two key reasons. Firstly, ER, specifically rehearsal-based methods, remains highly effective in mitigating forgetting, especially in the demanding Class-IL setting. Secondly, the adoption of ER aligns with the understanding of the human brain, where rehearsal is considered a critical component for information consolidation, contributing to lifelong learning.
>
> Our findings illustrate that Multimodal Experience Replay significantly enhances model performance in challenging settings. For instance, in the case of Seq-VGGSound, Multimodal ER achieves superior performance even with half the number of samples compared to unimodal CL (compare unimodal performance with a buffer size of 1000 with multimodal ER with a buffer size of 500). The performance gains are even more pronounced with our proposed structure-aware multimodal replay. These results underscore the effectiveness of multimodal learning and replay, which is a key takeaway from our study.
>
> Furthermore, while there are several state-of-the-art methods in unimodal CL, the majority of them cannot be applied directly in a multimodal setting and would require modifications and adaptation to multimodal architecture.
>
> Additionally, our ablation study in the Appendix (Table 3) delineates the contributions of various components in Structure-Aware Multimodal Learning. Notably, the combination of Unimodal Memory and Consistency Regularization (UM + CR)  corresponds to an adaptation of DER++ [1] in a multimodal setting, which serves as a strong baseline in unimodal CL. We demonstrate that SAMM provides substantial performance gains over this baseline (34.51 vs. 29.13).
>
> Importantly, the overarching message from our study is that akin to humans, multimodal learning may be pivotal in enabling effective CL in DNNs. We aim to encourage the broader research community to transition from unimodal CL to multimodal CL.
>
> >Could the authors further clarify why the proposed approach can not be applied to vision language? What is the advantage of the proposed benchmark compared with [3], besides the modality difference?
>
> Our study can indeed be applied to vision-language models and it would be interesting to extend our framework to additional modalities and we believe they will lead to even more robust and holistic representation learning. We chose Visual and Audio modalities as video is a more natural way to represent objects and actions, similar to how humans consume information in the real world. We hope that our work will inspire future work to integrate additional modalities.
>
> The main difference between our proposed benchmark and CLiMB [3] doesn’t lie in the modalities used but rather in the CL scenarios itself. CLiMB focuses on tasks that are inherently language and vision-based and by their very nature cannot be done in unimodal settings additionally, they focus on transfer learning in dataset incremental learning where the tasks share little to no similarity. Not only does this not allow us to see the benefits of multimodal CL over unimodal CL but also does not form a correspondence between the established challenging CL scenarios in an unimodal setting (Class-IL and Domain-IL). In contrast, our framework naturally extends the object recognition task to a multimodal setting (using videos which is similar to how humans solve the object/action recognition task). It also allows us to adapt the progress in unimodal CL to multimodal CL and can allow the research community to gauge the benefits of multimodal CL and more smoothly transition towards it.

---

> ### Author Response · Authors · 2023-11-21
> **Author's response (2/2)**
>
> >I understand that [4] is published after the submission ddl, but it would be good if the authors could comment a few sentences about the differences with them in the rebuttal.
>
> [4] can indeed be considered as parallel work which echoes a similar message as ours and we actually see it as a valuable work that supports our claim that multimodal CL is a promising approach to enabling CL in DNNs. Our proposed benchmark provides a more holistic evaluation than the evaluation in [4] as in addition to Class-IL we also simulate Domain-IL and Generalized Class-IL whereby the model has to tackle additional challenges of class imbalance and learn over recurring classes.

---

> ### Author Response · Authors · 2023-11-23
> **Activation Maps Analysis**
>
> >For the semantic-aware feature alignment, I wonder if the authors can provide some visualization examples to demonstrate that the model indeed learns the desired modality-specific features, such as Figure 4 in [5].
>
> We thank the reviewer for the very valuable suggestion. We performed this analysis and have added the results in the Activation Maps section in the Appendix.  The activation maps in Figure 6 show that MultiModal ER allows the model to attend to regions in the image that are associated with the label and localize the sound. SAMM considerably improves sound localization and attends to the most pertinent regions in the image. For playing saxophone, SAMM attends more to the saxophone and the mouth regions. Similarly, for tap dancing, SAMM rightfully attends more to the legs. This shows that multiple modalities and structure-aware alignment in SAMM enable the model to learn more holistic representations and focus on the regions associated with the class. The enhanced localization of sound in SAMM shows that our method effectively aligns the two modalities and learns a more holistic representation.

---

### Official Review · Reviewer_67Cp · 2023-11-05

**Soundness:** 2 fair
**Presentation:** 2 fair
**Contribution:** 3 good
**Rating:** 5
**Confidence:** 4

**Summary:**

The paper proposes a new benchmark based on the VGGSound dataset for multimodal (visual-audio) continual learning (CL).

The authors show complementary aspects with the results of analyses on the dataset to highlight the advantageous points of integrating multiple modalities of visual and audio.

Also, the paper presents a method for integrating and aligning information from multiple modalities using relational structural similarities, which seems to induce more robust representations to reduce catastrophic forgetting in deep neural networks.

**Strengths:**

- The authors introduce novel benchmark datasets for multimodal CL on vision and audio. If publicly available, it would be valuable and helpful for our communities to provide one of the standardized frameworks for evaluating the performance of models and facilitating fair comparisons between different methods in visual-audio multimodal CL settings.


- The paper presents empirical evidence supporting the complementary benefits of integrating multiple modalities of vision and audio. It seems to have better representations to be robust to reduce catastrophic forgetting.

**Weaknesses:**

- ﻿The paper shows the main experimental results of the proposed method, SAMM (Semantic-aware multimodal method), in Table 1~2. I think that the performances of other methods reported in major references such as [Buzzega et al., NIPS20] or [Arani et al., PAMI 2022] seem to be compared. Since lack of comparison, it is NOT clear to figure out the effectiveness and uniqueness of the proposed method among other methods.

- The paper does NOT provide enough information (including data composition, details on evaluation, and experimental settings) to reproduce the results in the experiments, even though Appendix A.2~A.4 presents some information.

- It seems weak as a paper to propose a new dataset. Because it needs to provide baseline performances to show the characteristics of the dataset. On the other hand, it seems weak as a paper to propose a novel method for continual learning for visual-audio multimodal settings since it does not clearly validate the pros and cons of the proposed method.




-- Minor
- 5th line on page 5, models a capture --> models to capture?
- caption in Figure 4, leverages leverage --> leverages?
- lines in Figure 3, it would be better to draw with line styles (solid, dotted, ... ). It is not easy to discriminate in gray-color printing.



-----------------------------------
Post-rebuttal
-----------------------------------
I've read the authors' feedback and other reviewers' opinions.
Even though the rebuttal addresses my major concerns, it partially resolves mine.
I'd like to keep my score as it is.

**Questions:**

- Is there any reason to compare with only ER?

- What is the motivation to introduce relational structural similarity into the proposed models?

---

> ### Author Response · Authors · 2023-11-21
> **Author's response (1/2)**
>
> We thank the reviewer for their positive feedback and valuable insights. We are glad that the reviewer sees value in integrating multiple modalities and the proposed multimodal benchmark for the community, and we do aim to make it publicly available for the research community.
>
> >The paper shows the main experimental results of the proposed method, SAMM (Semantic-aware multimodal method), in Table 1~2. I think that the performances of other methods reported in major references such as [Buzzega et al., NIPS20] or [Arani et al., PAMI 2022] seem to be compared. Since lack of comparison, it is NOT clear to figure out the effectiveness and uniqueness of the proposed method among other methods.
>
>
> We would like to emphasize that the main goal of our study is not necessarily to achieve state-of-the-art results but rather to advocate for the adoption of multimodal continuous learning (CL) and to establish a standardized benchmark setting for research in this area. Therefore, the focus of our work was on demonstrating the advantages of multimodal experience replay over unimodal experience replay in challenging CL scenarios.
>
> Additionally, we propose a promising direction for aligning modalities to enhance complementary learning. This alignment is proposed in a manner that not only enables multimodal inference but also serves as a baseline for unimodal inference, thereby creating a foundation for future research to build upon.
>
> Furthermore, while there are several state-of-the-art methods in unimodal CL, the majority of them cannot be applied directly in a multimodal setting and would require modifications and adaptation to multimodal architecture.
>
> However, the ablation study in Appendix (Table 3) shows the contributions of the different components in SAMM. *UM + CR* would correspond to an adaptation of DER++ [Buzzega et al., NIPS20] in a multimodal setting, and we show that SAMM provides considerable performance gains over it (34.51 vs. 29.13).
>
> > The paper does NOT provide enough information (including data composition, details on evaluation, and experimental settings) to reproduce the results in the experiments, even though Appendix A.2~A.4 presents some information.
>
> We have added further details about the dataset composition, and experimental setup to enhance the reproducibility of our results. Additionally, we will make the code and dataset publicly available.
>
> > It seems weak as a paper to propose a new dataset. Because it needs to provide baseline performances to show the characteristics of the dataset. On the other hand, it seems weak as a paper to propose a novel method for continual learning for visual-audio multimodal settings since it does not clearly validate the pros and cons of the proposed method.
>
> Our study aims to make a case for multimodal learning, as multisensory information processing is one of the salient features of the human brain, which may account for its CL capabilities. Not only do multiple modalities enable the model to learn a holistic and robust representation of objects, but multi-modal replay may also allow for better knowledge consolidation and less forgetting. Therefore, the key takeaway message from our study is that, similar to humans, multimodal learning might be critical in enabling effective CL in DNNs, and we want to encourage the research community at large to shift towards multimodal CL from unimodal CL.
>
> We welcome any suggestion or recommendation from the reviewer to more clearly validate the pros and cons of the proposed approach.

---

> ### Author Response · Authors · 2023-11-21
> **Author's response (2/2)**
>
> >Is there any reason to compare with only ER?
>
> While there are other approaches to CL including regularization-based methods and architecture-based methods and it would be interesting to see how they can be adapted in a multimodal setting, we chose rehearsal-based methods for two specific reasons. Firstly, these methods have consistently demonstrated high efficacy in addressing the challenge of forgetting, especially in the demanding Class-IL setting. Traditional regularization-based approaches falter in this context, and many dynamic architecture-based methods are either tailored for task-IL, which doesn't align with the requirements of Continual Learning (CL), or they scale linearly with the number of tasks. Secondly, the incorporation of rehearsal aligns with the understanding of the human brain, where rehearsal is considered a critical component for information consolidation [1], contributing to the facilitation of lifelong learning.
>
> > What is the motivation to introduce relational structural similarity into the proposed models?
>
> The main motivation for introducing relational structural similarity was to better align the different modalities in a manner that doesn’t restrict them from learning modality-specific optimal representations. The key premise is that learning representations in each modality that retain the relational structure between data points would enhance the alignment between the different modalities, which is one of the biggest challenges in multimodal learning. This allows the model to learn better joint representations that can leverage complementary information and learn a more holistic and robust representation of data that is less vulnerable to forgetting.
>
> References:
>
> [1] McClelland, James L., Bruce L. McNaughton, and Randall C. O'Reilly. "Why there are complementary learning systems in the hippocampus and neocortex: insights from the successes and failures of connectionist models of learning and memory." Psychological review 102.3 (1995): 419.

---

### Official Review · Reviewer_i2PA · 2023-11-06

**Soundness:** 2 fair
**Presentation:** 3 good
**Contribution:** 2 fair
**Rating:** 3
**Confidence:** 5

**Summary:**

This paper proposes a multi-modal continual learning benchmark.  Further, this paper also provides a simple baseline by incorporating the knowledge contained in different modalities to achieve better multi-modal continual learning with less forgetting on previously learned tasks. Experiments on a visual and audio modality continual learning dataset show the effectiveness of the proposed method compared to standard experience replay.

**Strengths:**

* The paper is easy to follow.

* This paper provides a baseline of multi-modal continual learning and benchmark.

**Weaknesses:**

* The proposed method is straightforward with experience replay and those techniques are commonly used in existing multimodal learning and continual learning literature.


* The memory buffer includes multi-modal examples from previous tasks.  The authors store the same number of data for single-modality and multi-modality. It would be better to compare different modality methods in terms of the same memory storage since multi-modality memory data requires more storage to store multi-modality data.


* The baseline is too weak, only the standard experience replay is compared. It would be better to compare to more recent state-of-art baselines in experience replay.


* Furthermore, there are other categories of CL methods, including regularization-based methods and architecture-based methods. It would be better to also compare those methods in the experiment.


* The experiments are only performed on visual and audio modality. It would be better to provide experiment and benchmark on other modalities as well, e.g., language.

**Questions:**

N/A

---

> ### Author Response · Authors · 2023-11-21
> **Author's response (1/2)**
>
> We thank the reviewer for the feedback and hope to address their concern below:
>
> >The proposed method is straightforward with experience replay and those techniques are commonly used in existing multimodal learning and continual learning literature.
>
> Could the reviewer please share the existing multimodal continual learning literature that applies these techniques in multimodal class-IL settings? We are not aware of earlier literature that has studied the effect of multiple modalities in our proposed benchmarking settings with parallels with single-modality learning. The main contribution of our study is to bring more attention to multimodal CL and to showcase the benefits of multimodal CL over unimodal CL through an extensive empirical study. To further encourage research in this promising direction and facilitate standardization similar to the unimodal setting, we present a multimodal CL benchmark that simulates the challenges in unimodal CL to enable comparison. Additionally, we provide a baseline that provides a promising approach for leveraging complementary information in the different modalities.
>
> >The memory buffer includes multi-modal examples from previous tasks. The authors store the same number of data for single-modality and multi-modality. It would be better to compare different modality methods in terms of the same memory storage since multi-modality memory data requires more storage to store multi-modality data.
>
> We believe that the constant number of experiences is a good comparison to show how well a multimodal modal can retain knowledge given the same number of data samples as they provide richer and complementary information. Also, it is worth noting that audio samples consume much less memory compared to video samples, so the memory utilization for audio video (multimodal) is comparable to the memory for video only. Furthermore, it is more valuable to study the value of data samples.
>
> Importantly, our study aims to make a case for multimodal learning as multisensory information processing is one of the salient features of the human brain which may account for its CL capabilities. Not only do multiple modalities enable the model to learn a holistic and robust representation of objects but multi-modal replay may also allow for better knowledge consolidation as less forgetting. Therefore, to assess the richness and effectiveness of experience replay in a multimodal setting, it is only fair to see the performance gain with the same number of experiences.
>
> However, to alleviate the concern of the reviewer further, we ask them to compare the results of Multimodal replay with 1000 buffer size in Seq-VGGSound with the results of Audio and Video with double the buffer size. Multimodal replay achieves 28.09 with 1000 buffer size which the unimodal counterparts fail to surpass even with 1000 more replay samples (23.41 with Audio and 14.19 with Video). The same holds for comparison between 500 samples for multimodal replay and 1000 samples for unimodal replay. The performance gains and difference is even more pronounced with our proposed structure-aware multimodal replay. These results show the effectiveness of multimodal replay.
>
>
> >The baseline is too weak, only the standard experience replay is compared. It would be better to compare to more recent state-of-art baselines in experience replay.
>
> We would like to emphasize that the goal of our study was not to propose a state-of-the-art but rather to provide a convincing case for moving towards multimodal CL and provide a standardized multimodal CL benchmark setting to facilitate research in this promising direction. Showcasing the performance gains of multimodal experience replay compared to unimodal experience replay in challenging CL settings (corresponding to the three established CL settings in unimodal settings) is the main contribution of our work. We additionally provide a promising direction for aligning the modalities to enhance complementary learning in a manner that allows multimodal as well as unimodal inference as an initial baseline upon which future work can build.
>
> Furthermore, while there are several state-of-the-art methods in unimodal CL, the majority of them cannot be applied directly in a multimodal setting and would require modifications and adaptation to multimodal architecture.
>
> Importantly, the key takeaway message from our study is that, similar to humans, multimodal learning might be critical in enabling effective CL in DNNs, and we want to encourage the research community at large to shift towards multimodal CL from unimodal CL.

---

> ### Author Response · Authors · 2023-11-21
> **Author's response (2/2)**
>
> >Furthermore, there are other categories of CL methods, including regularization-based methods and architecture-based methods. It would be better to also compare those methods in the experiment.
>
> We agree with the reviewer that there are other approaches to CL including regularization-based methods and architecture-based methods and it would be interesting to see how they can be adapted in a multimodal setting. We opted for rehearsal-based in our study owing to two reasons: 1) Rehearsal-based methods continue to be the most effective in mitigating forgetting particularly in the challenging Class-IL setting where regularization-based approaches fail and the majority of dynamic architecture-based methods are either designed for task-il which doesn’t meet the desiredetas of CL [1] or scale linearly with the number of tasks. 2) The human brain is believed to employ rehearsal as one of the critical components for consolidating information [2] and enabling lifelong learning.
>
> The performance gains with Multimodal CL with experience replay in challenging CL settings provide a strong case for moving towards multimodal CL and designing approaches tailored for effectively utilizing multiple modalities to learn robust representations that are less vulnerable to forgetting.
>
> > The experiments are only performed on visual and audio modalities. It would be better to provide experiment and benchmark on other modalities as well, e.g., language
>
> It would indeed be interesting to extend our framework to additional modalities and we believe they will lead to an even more robust and holistic representation of learning. We chose Visual and Audio modalities as video is a more natural way to represent objects and actions, similar to how humans consume information in the real world. We hope that our work will inspire future work to integrate additional modalities.
>
> References:
>
> [1] Farquhar, Sebastian, and Yarin Gal. "Towards robust evaluations of continual learning." arXiv preprint arXiv:1805.09733 (2018).
>
> [2] McClelland, James L., Bruce L. McNaughton, and Randall C. O'Reilly. "Why there are complementary learning systems in the hippocampus and neocortex: insights from the successes and failures of connectionist models of learning and memory." Psychological review 102.3 (1995): 419.

---

### Author Response · Authors · 2023-11-23
**General Comment**

We thank all the reviewers for their feedback and valuable suggestions. We are glad that the reviewers see the value in the proposed benchmark and the effectiveness of multimodal replay in CL. We have attempted to address all the concerns of the reviewers in our individual responses. Based on the reviewer's suggestions, we have made the following changes (indicated by blue color in the paper) in the revised version:
- **Additional details about the experimental setup and dataset** to improve reproducibility. Note that for further reproducibility and to make the multimodal benchmark available to the research community at large, the dataset files and code for the method and benchmark will be made publicly available upon acceptance.
- **Analysis of the activation maps** in Appendix. The activation maps in Figure 6 show that MultiModal ER allows the model to attend to regions in the image that are associated with the label and localize the sound. SAMM considerably improves sound localization and attends to the most pertinent regions in the image. This shows that multiple modalities and structure-aware alignment in SAMM enable the model to effectively align the two modalities and learn a more holistic representation.

We would also like to reiterate the primary objective of our study was to study the role of multiple modalities in enabling CL and to present a compelling case for moving towards Multimodal CL. Our study is based on the premise that the human brain's capacity for processing multisensory information is a pivotal factor contributing to its continual learning capabilities. Our empirical findings strongly support the transition to MultiModal CL. The results underscore that leveraging multiple modalities enables the model to learn more holistic and robust representations of objects and actions. Moreover, our study suggests that employing multimodal replay improves knowledge consolidation, and mitigates forgetting.

It is crucial to emphasize that our intention was not to present a state-of-the-art model. Instead, we aimed to build a compelling case for embracing multimodal CL, as well as offering a standardized benchmark setting to facilitate research in this promising direction. The main contribution of our work lies in demonstrating the performance gains of multimodal learning and experience replay over unimodal experience replay in challenging CL scenarios, mirroring the established CL settings in unimodal contexts. Additionally, we propose a promising direction for aligning modalities to enhance complementary learning, providing a strong baseline for future work. Furthermore, while there are numerous state-of-the-art methods in unimodal CL, it is essential to note that the majority necessitate modifications to be applicable in a multimodal context.

Our study urges the research community to recognize the significance of multimodal CL, emphasizing its potential as a critical element in achieving effective CL in DNNs. We encourage a collective shift from unimodal CL to multimodal CL, drawing inspiration from the parallel in human learning processes.

---

### Meta-Review · Area_Chair_Vbqn · 2023-12-18

**Metareview:**

This paper addresses catastrophic forgetting in continual learning, and studies interactions of multiple modalities, vision and audio, in mitigating forgetting. It introduces a benchmark for multi-modal continual learning, and extends an experience replay approach to multimodal continual learning.

While the reviewers acknowledged the importance of the benchmark dataset for multimodal
continual learning, they raised several important concerns:
(1) insufficient experiments and baseline comparisons to assess significance of the benchmark as well as the efficacy of the proposed approach of multimodal experience replay – unanimously all reviewers, see the comments; (2) lack of clarity on the main technical contribution – see Reviewer 67Cp comments, (3) lack of clarity in the experiments – see Reviewer 67Cp comments, (4) lack of language modality in the benchmark for multi-modal continual learning and memory storage analysis – see Reviewer i2PA concerns.

The rebuttal was able to clarify some questions, but did not manage to sway any of the reviewers. A general consensus among reviewers and AC was reached to reject the paper. Ultimately, the weaknesses in the evaluation of the benchmark baselines (1) were just too hard to overlook. We hope the reviews are useful for improving and revising the paper.

**Justification For Why Not Higher Score:**

All reviewers and AC are in consensus about rejection.

**Justification For Why Not Lower Score:**

N/A

---

### Decision · Program_Chairs · 2024-01-16

Reject